# Recalibrating vision-for-action requires years after sight restoration from congenital cataracts

Irene Senna[1,2]*[†], Sophia Piller[1], Itay Ben-Zion[3], Marc O Ernst[1]*

[1]Applied Cognitive Psychology, Faculty for Computer Science, Engineering, and Psychology, Ulm University, Ulm, Germany; [2]Liverpool Hope University, Liverpool, United Kingdom; [3]Pediatric Ophthalmology Service, Padeh Medical Center, Tiberias, Israel

**Abstract** Being able to perform adept goal-directed actions requires predictive, feed-forward control, including a mapping between the visually estimated target locations and the motor commands reaching for them. When the mapping is perturbed, e.g., due to muscle fatigue or optical distortions, we are quickly able to recalibrate the sensorimotor system to update this mapping. Here, we investigated whether early visual and visuomotor experience is essential for developing sensorimotor recalibration. To this end, we assessed young individuals deprived of pattern vision due to dense congenital bilateral cataracts who were surgically treated for sight restoration only years after birth. We compared their recalibration performance to such distortion to that of age-matched sighted controls. Their sensorimotor recalibration performance was impaired right after surgery. This finding cannot be explained by their still lower visual acuity alone, since blurring vision in controls to a matching degree did not lead to comparable behavior. Nevertheless, the recalibration ability of cataract-treated participants gradually improved with time after surgery. Thus, the lack of early pattern vision affects visuomotor recalibration. However, this ability is not lost but slowly develops after sight restoration, highlighting the importance of sensorimotor experience gained late in life.

## Editor's evaluation

This valuable paper will be of interest to researchers in the fields of motor control, visual perception, learning and brain plasticity, sight loss, and rehabilitation. The paper shows the contributions of sensory-motor experience to the development of visuo-motor recalibration abilities using convincing and careful experimental methods and analyses, comparing a rare population of late-operated cataract patients with normal-sighted control groups.

## Introduction

Most of our visually controlled actions, such as grasping objects, proficiently walking to target locations, effortlessly making use of tools or alike, are skillful and adept. Such a fast and efficient behavior is difficult to achieve relying solely on feedback control, because sensory feedback during movements is typically delayed and would require constant monitoring. Hence, to achieve such proficiency, we predominantly rely on feedforward control. Feedforward control avoids the use of online feedback (visual and/or proprioceptive) by including accurate model predictions. For example, successfully reaching for targets based on predictive feedforward control requires a model including the mapping between the visually estimated target location and the motor commands necessary to reach for it

**\*For correspondence:**
sennai@hope.ac.uk (IS);
marc.ernst@uni-ulm.de (MOE)

**Present address:** [†]Department of Psychology, Liverpool Hope University, Liverpool, United Kingdom

**Competing interest:** The authors declare that no competing interests exist.

(*Wolpert and Ghahramani, 2000*; *Wolpert et al., 2011*). Acquiring such a model needs plenty of experience and thus time. As the state of our body, the range of tools we use, or the visual input constantly change, this sensorimotor mapping requires constant updating, known as recalibration (*Burge et al., 2008*; *Redding et al., 2005*; *von Helmholtz, 1867*). Typically, humans are able to effectively recalibrate their sensorimotor system, which is the basis for the many visuomotor skills we perform. The first signs of the ability to recalibrate emerge in the very first weeks of life and such ability keeps sharpening during childhood (*Contreras-Vidal et al., 2005*; *Ferrel et al., 2001*; *Gómez-Moya et al., 2016*; *McDonnell and Abraham, 1979*; *McDonnell and Abraham, 1981*; *Riddell et al., 1999*). Here, we ask whether the ability to recalibrate the visuomotor system requires visual and visuo-motor experience early in life to develop.

To this end, we tested a group of children and adolescents who did not have pattern vision for the first years of life due to dense congenital bilateral cataract, and therefore, had no chance to perform coordinated visually controlled actions in a skillful fashion. We investigated whether they would show the ability to recalibrate their visuomotor system immediately after the recovery of pattern vision following cataract removal surgery and, if not, whether such an ability could still be acquired with experience in the months to years after surgery.

To study the development of the ability to recalibrate, we experimentally introduced perturbations to the visual input using prism goggles that shifted the visual field laterally. The recalibration performance was then assessed using a rapid pointing task. Usually, when typically sighted adults are exposed to such visual shifts, they initially show a deviation in their pointing movements in the same direction as the visual perturbation (systematic error) (e.g. *Burge et al., 2008*; *Redding et al., 2005*). However, this systematic error is quickly reduced by gradually updating the pointing behavior. That is, after only a few pointing movements while wearing such prisms, sighted individuals are able to correctly point to the target location again, indicating that they have adapted to the visual distortion. Furthermore, an additional signature of sensorimotor recalibration is the occurrence of an aftereffect upon removal of the prism distortion, i.e., a systematic displacement of the pointing movement in the opposite direction to the visual distortion. Here we tested whether cataract-treated individuals are able to recalibrate their visuomotor system, and found that it takes them months to years to reach a performance level comparable to that of sighted individuals.

## Results
### Recalibration in cataract-treated participants and sighted controls

Participants were asked to perform a pointing task consisting of three phases (*Fortis et al., 2010*; *Frassinetti et al., 2002*). First, we determined their baseline performance (*pre-prism* phase): to this end, individuals were asked to repeatedly point toward a visual target while wearing neutral goggles that did not introduce any visual distortion. The pointing movements were performed in the absence of any visual feedback of the arm, which was occluded by the experimental setup (see Materials and Methods). Next, in the *prism* phase, participants executed the task while wearing prism goggles shifting the apparent target location visually to the right by 20 prism diopters (11.31°). In this phase, participants could see the tip of their index finger at the end of the pointing movement appearing from below the setup (terminal feedback), while the rest of the arm movement was hidden from view to prevent any online corrections during the pointing movement (*Figure 1A*, upper panel). This terminal feedback of the pointing error is generally sufficient for the sighted population to recalibrate their visuomotor system from trial to trial (*Burge et al., 2008*; *Redding et al., 2005*; *von Helmholtz, 1867*). Finally, the distortion prisms were removed and participants were tested again with the neutral goggles in the absence of terminal visual feedback (*post-prism* phase). In each trial, pointing errors were measured as the difference–in degrees of visual angle–between the target location and the pointing endpoint along the horizontal axis.

We compared the performance of a group of 20 Ethiopian children and adolescents suffering from congenital dense bilateral cataracts, surgically treated 5–19 years after birth and tested days to years after surgery, with that of two control groups (*Figure 1—source data 1*). The first control group consisted of 20 typically developing sighted participants who were individually matched for age with the cataract-treated sample, as we found that age has an influence on the recalibration rate in the healthy population (*Figure 1—figure supplement 6*). In the second control group, 20 sighted

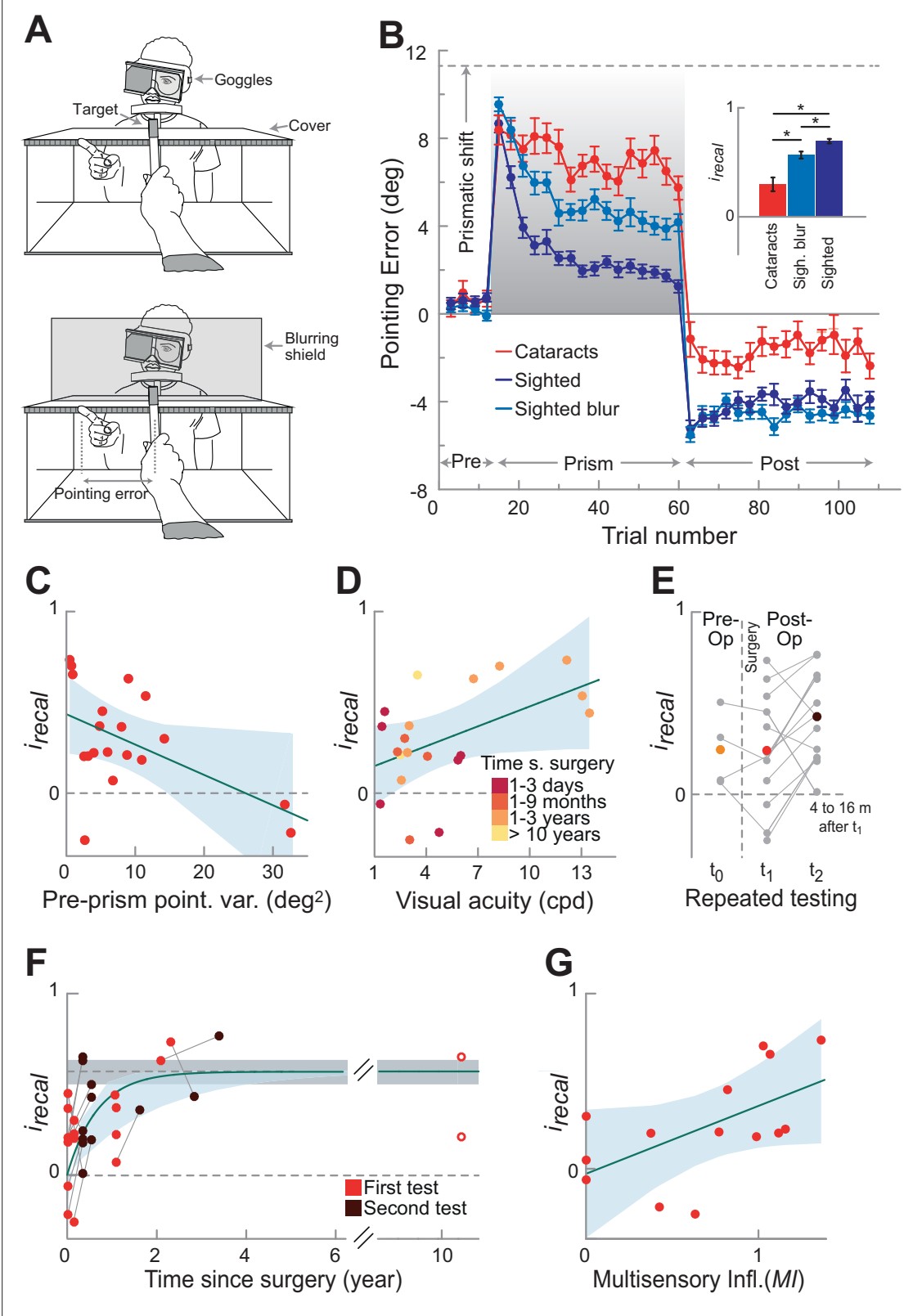

**Figure 1.** Recalibration behavior. (**A**) Pointing setup. Upper panel without, lower panel with blurring shield. Participants wore goggles with an eye cover over the non-dominant eye. A prism could be inserted into the goggles during the *prism* phase inducing a rightward shift. Participants performed pointing movements toward a target. A cover was used to block vision during pointing movement. Only during the *prism* phase participants could see their terminal pointing error, by seeing the tip of their finger appearing from under the cover. Before (*pre-prism*) and after (*post-prism*) the *prism* phase,

*Figure 1 continued on next page*

*Figure 1 continued*

participants performed the same task without the prism and in the absence of terminal feedback. To mimic the poor visual acuity of the cataract-treated participants, we blurred vision in a group of sighted control participants by placing a blurring shield in front of the target (lower panel). (**B**) Recalibration Performance. Mean pointing errors across bins of three trials are calculated for each participant across the three phases of the experiment (*prism* phase in gray), and group averages are shown for the three groups: cataract-treated (red), and sighted controls tested with (light blue) and without (blue) visual blur (n=20 in each group). The dashed line represents the prismatic shift (11.31°). Error bars show SEM across participants. The inset shows the recalibration index $i_{recal}$, which summarizes the recalibration performance in the *prism* phase and in the first three trials of the *post-prism* phases (0 no recalibration, 1 complete recalibration). The analysis on $i_{recal}$ showed that each group differed from the other (Bonferroni corrected Wilcoxon rank-sum, all p≤0.006, following Kruskal–Wallis test, $\chi^2_{(2)}$=27, p<0.0001, $\eta^2$=0.38). Although the cataract-treated group recalibrated less than the sighted control groups tested with and without visual blur ($i_{recal}$, mean ± SEM = 0.30 ± 0.06, 0.57 ± 0.03, and 0.69 ± 0.02, respectively), their recalibration performance was significantly greater than 0 (Wilcoxon signed-rank test, z=3.25, p=0.0012). (**C**) Relation between recalibration performance $i_{recal}$ and pointing precision during *pre-prism* phase (baseline) in cataract-treated participants (n=19, one outlier above three SD from the mean was excluded). The variance in the pointing errors at baseline negatively correlated with $i_{recal}$ (Pearson's correlation coefficient, r=−0.53, p=0.019; if we include also the outlier, r=−0.46, p=0.040), showing that participants with noisier performance at baseline recalibrate less. The light-blue shaded area indicated 95% confidence intervals of the regression line. (**D**) Recalibration performance $i_{recal}$ as a function of visual acuity in cataract-treated participants (n=20). Participants with higher visual acuity recalibrate more (higher $i_{recal}$, Pearson's correlation coefficient, r=0.5, p=0.025). The data points are coloured with brighter colors indicating longer time since surgery. This shows that individuals tested soon after surgery tended to recalibrate less (smaller $i_{recal}$). Note, however, that time since surgery did not significantly correlate with visual acuity at the group level (r=−0.03, p=0.8). (**E**) Repeated testing in a subset of 13 participants tested over time. Four of them were tested also before surgery (since their visual acuity allowed them to see the target). Although their CSF increased due to surgery, their degree of recalibration did not in the first tests performed a few days after surgery. Instead, with more time after surgery (4–16 months) their recalibration performance significantly improved (Wilcoxon signed-rank test, z=20, p=0.04, one-tailed). Individual data are reported in gray, with connecting lines linking the same participant, while mean performance for each testing time is reported in larger filled colored circles. (**F**) Recalibration performance as a function of time since surgery. After surgery participants tended to exponentially improve their recalibration performance $i_{recal}$ (time constant, b=1.5, CI=[0.51, 2.49]), reaching the level of the CSF-matched controls (i.e. tested with visual blur) at around 2 years (however, due to the large inter-subject variability, this estimate contains substantial uncertainty). Note that the exponential fit is not driven by the two participants tested more than 10 years after surgery: when excluding them from the exponential fit, the time constant b (b=1.5, CI=[0.39, 2.67]) is comparable to the one obtained in the whole sample. The dashed line and the gray shaded area indicate the mean performance and the 95% confidence intervals of the sighted CSF-matched controls, respectively. The exponential (dark green curve, with the ligth-blue shaded area indicating its 95% confidence intervals) is fitted on all measurements obtained from the participants after surgery, with red circles indicating the first post-surgical test and brown circles indicating the second performance of the subset of 13 participants re-tested in the same task (**E**), with connecting lines linking the same participant. (**G**) Correlation between recalibration and multisensory integration. Fourteen participants took part in this and in a previous study on multisensory integration (*Senna et al., 2021*) at around the same time after surgery. We investigated the relationship between the performance in both tasks by correlating $i_{recal}$, as a measure of recalibration performance and the Multisensory Influence (*MI*) as a measure of integration performance between vision and touch: r=0.58, p=0.03.

The online version of this article includes the following source data and figure supplement(s) for figure 1:

**Source data 1.** Clinical characteristics of the cataract-treated participants.

**Figure supplement 1.** Contrast sensitivity functions in cataract-treated participants and in sighted controls tested with visual blur.

**Figure supplement 2.** Accuracy and precision in the pre-prism phase.

**Figure supplement 3.** Pointing responses as a function of target location in the cataract-treated participants.

**Figure supplement 4.** Development of the aftereffect throughout the *post-prism* phase.

**Figure supplement 5.** Correlational analyses, separated for adaptation and aftereffect, in the cataract-treated group.

**Figure supplement 6.** Developmental path of recalibration in typically sighted individuals.

participants were individually matched to cataract-treated individuals for both age and visual acuity, using visual blur filters to degrade vision. This was necessary because previous research showed that visual uncertainty in the form of blur can potentially result in a decreased learning rate (*Burge et al., 2008*). Thus, this second control group provides a baseline for the effect of visual acuity on adaptation rate. To this end, we determined the cutoff frequency of the contrast sensitivity function (CSF) in each cataract-treated participant (*McKyton et al., 2015*; *Senna et al., 2021*). We then blurred vision in each control by placing a blurring filter between the participant and the visual target until the desired shape and cutoff frequency of the CSF were matched (*Figure 1A*, lower panel, Materials and Methods, and *Figure 1—figure supplement 1*).

We first verified that the cataract-treated participants were able to accurately point to all target locations before being exposed to the prism distortion (*Figure 1—figure supplements 2 and 3*). Their pointing accuracy in the *pre-prism* phase did not significantly differ from that of the two control groups (Kruskal–Wallis test on pointing errors, $\chi^2_{(2)}$=1, p=0.61, $\eta^2$=0.017), while their pointing

precision was significantly reduced compared to the control groups tested either with or without visual blur (Bonferroni-corrected two-sample F-tests on the variance of the pointing errors, $F_{(19,19)}$=45.00, p<0.0001 and $F_{(19,19)}$=95.98, p<0.0001, respectively). Next, to quantify the adaptation rate for the different groups in the *prism* phase, we fitted power functions to the trial-by-trial pointing errors averaged across the participants of each group (see Material and methods):

$$Error = a * x^b. \tag{1}$$

The parameter *a* is the amplitude of pointing error and *b* is the time constant, that is the adaptation rate: the larger *b*, the faster the adaptation to the visual distortion. This analysis showed that cataract-treated individuals have difficulties recalibrating their sensorimotor system: compared to sighted controls, the group of cataract-treated participants only marginally reduced their pointing error, although the learning was still significant (i.e. the 95% confidence interval for the adaptation rate *b*=−0.09 CI=[-0.14,–0.03] did not include 0). Instead, sighted participants tested in normal visual conditions had, on average, the highest learning rate *b*=−0.49 (CI=[–0.54,–0.45]) of all groups tested. As expected (*Burge et al., 2008*), blurring vision in sighted controls (thus matching them in CSF to the cataract participants) resulted in a reduced adaptation rate (*b*=−0.27, CI=[–0.32,–0.22]). Nonetheless, they were still faster at adapting to the visual distortion than the cataract-treated individuals, as shown by the fact that they presented a larger *b*, with CI that did not overlap with those of the cataract-treated group (*Figure 1B*).

Notably, all groups showed an aftereffect: after prism removal in the *post-prism* phase, all groups presented a pointing bias, i.e., a systematic error in the opposite direction to the displacement induced by the prism. The mean aftereffect across all trials of the *post-prism* phase displayed by cataract-treated participants was less pronounced than that of both control groups (around –2° vs –4.5°, respectively; cataract-treated vs sighted with and without visual blur, Bonferroni-correctedWilcoxon rank-sum test: z=3.23, p=0.0036 and z=3.26, p=0.0033, respectively, following Kruskal–Wallis test: $\chi^2_{(2)}$=14.3, p=0.0008 $\eta^2$=0.221; Materials and methods), and did not show an analogous trial-by-trial decay over time (*Figure 1—figure supplement 4*). Importantly, however, the aftereffect observed in cataract-treated participants significantly differed from zero (Wilcoxon signed-rank test, z=2.31, p=0.02), confirming the results from the *prism* phase that at the group level they also showed some visuomotor recalibration performance (*Figure 1B*). In a further step, we investigated which factors could have contributed to the recalibration performance of the cataract-treated participants at the individual level. To this end, we first calculated a recalibration index ($i_{recal}$) for each participant, which combined the amount of error reduction in the *prism* phase (*Adaptation*) and the systematic error right after prism removal (*Initial aftereffect*) into one index, therefore increasing power. The index was calculated by taking the average between *Adaptation* (i.e. induced *Prism Distortion*, 11.31°, minus the mean of the last three pointing errors of the *prism* phase) and the magnitude of the *Initial Aftereffect* (i.e. mean systematic error in the first three trials of the *post-prism* phase), normalized by the *Prism Distortion (11.31°, Equations 3; 4* in Materials and methods). Therefore, the index ranges from 0 (no recalibration) to 1 (complete recalibration, *Figure 1B*, inset). The two components of this index (*Adaptation* and *Initial Aftereffect*) were significantly correlated across all participants (Pearson *r*=0.39, p=0.002). Note that this index includes only the first trials of the *aftereffect*, thus it does not focus on the development of the *aftereffect*, which mainly involves proprioception, as it develops in the absence of terminal visual feedback (see Material and methods for more details, and *Figure 1—figure supplements 4 and 5* for analyses on the aftereffect throughout the entire *post-prism* phase).

## The contribution of sensory and sensorimotor uncertainty to recalibration

As increased uncertainty has been shown to affect the ability to recalibrate to visual distortions (*Burge et al., 2008*), we investigated whether there is a relationship between the precision of pointing at baseline (as a result of uncertainty) and the recalibration performance (as measured by $i_{recal}$) in cataract-treated participants. We found that the variance in the pointing errors in the *pre-prism* phase (as a measure of pointing precision) negatively correlated with $i_{recal}$ (Pearson's correlation coefficient, *r*=−0.53, p=0.019, *Figure 1C*). This finding indicates that participants presenting higher noise in their pointing performance (thus, showing higher noise in their sensorimotor system already at baseline) recalibrate less. In line with this finding, we also observed a significant correlation between visual

acuity and $i_{recal}$ ($r$=0.5, p=0.025, *Figure 1D*), meaning that higher visual uncertainty (leading to less precise pointing movements) results in poorer recalibration performance (*Burge et al., 2008*).

## The contribution of time after surgery to recalibration

In the typically sighted population, we found that $i_{recal}$ increases with age and thus experience (*Figure 1—figure supplement 6*), and it is already greater in the youngest control children (6–7 year-old, n=11, $i_{recal}$ = 0.62 ± 0.07) than in the whole cataract-treated sample (0.30 ± 0.06, Wilcoxon rank-sum, z=3.08, p=0.002, see *Figure 1—figure supplement 6*). The fact that much younger controls already recalibrate faster than cataract-treated participants may well indicate that learning to recalibrate is not merely an effect of brain maturation related to age, but requires experience to develop. We explored the possible contribution of time (and thus possibly experience) after surgery to the development of recalibration by re-testing a sub-group of 13 cataract-treated participants in a follow-up study. Participants of this subset were tested for the first time a few months after surgery (range: 1 day to 2 years 4 months, see *Figure 1—source data 1*) and retested 4–16 months later. Among them, a subset of 4 participants could be even assessed right before surgery. This was possible because this small subsample presented enough residual vision to be able to point to the targets and to have their CSF measured already prior to surgery. Despite a substantial improvement of their visual acuity right after cataract removal (CSF mean ± SEM pre: 2.14 ± 0.43; post: 4.53 ± 1), their mean recalibration performance did not improve accordingly, but stayed essentially the same between the pre-surgery assessment and the post-surgery evaluation occurring just a few days after surgery ($i_{recal}$ = 0.24 ± 0.11 vs 0.16 ± 0.14, respectively). The 13 cataract-treated individuals who were retested again several months after surgery showed a significantly higher $i_{recal}$ in the second (0.41 ± 0.07) as compared to the first post-surgical test (0.24 ± 0.09, Wilcoxon signed-rank test, z=20, p=0.04, one-tailed, *Figure 1E*). Instead, their visual acuity did not significantly differ between the two post-surgical tests (ΔCSF:1.2 ± 0.67; $t_{12}$=1.35, p=0.2). Importantly, their increased $i_{recal}$ could not simply be ascribed to a general reduction of the noise in their sensorimotor system, as the variance of the pointing errors in the *pre-prism* phase was comparable for the two tests (first: 10.05 ± 2.87 deg$^2$, second:10.34 ± 2.4 deg$^2$). Moreover, the variance in the second test was not correlated with $i_{recal}$ in the second test ($r$=0.24, p=0.42). Thus, the improvement in recalibration performance with time since surgery observed in the whole group of cataract-treated participants and the improvement observed in this specific subset tested twice cannot be explained only by a general reduction of the sensorimotor noise, already present at baseline.

Given that the ability to recalibrate appeared to improve over time, we ran a further analysis to estimate whether and in which timeframe cataract-treated participants might approach the performance level of the sighted controls. To this end, we included all measurements obtained from the participants after surgery (i.e. including also the second test of the subset of 13 individuals). As learning typically follows an exponential function rather than a linear trend, we fitted the time-since-surgery data using:

$$i_{recal} = (a - c) \, exp^{-bx} + c. \tag{2}$$

The asymptote $c$=0.57 was fixed to the mean performance of the CSF-matched control participants. The amplitude $a$ was fixed to 0, meaning that we assumed that without visual experience the recalibration performance corresponds to $i_{recal}$ = 0. From the fit we determined the time constant $b$ for the development of recalibration. $i_{recal}$ significantly improved with time since surgery and approached the asymptote, showing a performance indistinguishable from the matched controls, at around 2 years after surgery ($b$=1.5, CI=[0.51, 2.49], *Figure 1F*). However, given the high inter-subject variability in the recalibration ability of the cataract-treated participants, such an estimate will contain a large error margin.

## Relations between recalibration and multisensory integration

In a recent study, we found that cataract-treated participants learn to integrate visual and haptic information in a similar time frame following surgery as the one found here for the development of recalibration (*Senna et al., 2021*). Both abilities (multisensory integration and recalibration) pose a similar challenge, namely establishing correspondences between different sensory maps (for multisensory integration) and between sensory (e.g. vision and proprioception) and sensorimotor spaces. We investigated the possible relation between the two abilities in a subset of 14 participants that were tested

at around the same time following surgery in both studies (this one and the one from *Senna et al., 2021*). We found that $i_{recal}$ correlated with the *Multisensory Influence (MI)*, a measure of multisensory integration (see Materials and methods), showing that participants who were better at combining vision and haptics also had a better recalibration performance ($r$=0.58, p=0.03, *Figure 1G*). This result suggests that the two abilities might be related, and that what develops with time after surgery is the ability to establish correspondence between the sensory and the sensorimotor maps.

## Discussion

Here we investigated whether individuals suffering from congenital dense bilateral cataracts, surgically treated years after birth, can develop the ability for visuomotor recalibration. We used prism goggles to distort the visuomotor mapping and compared their recalibration performance to that of typically sighted individuals matched for age and visual acuity. Unlike typically developing individuals, who quickly recalibrated when exposed to distortions in the visuomotor mapping, we found that the recalibration ability of the cataract-treated individuals was almost absent right after surgery. Importantly, as time progressed after surgery they improved and started to show better recalibration performance, taking several months to years for developing to levels comparable with sighted individuals.

It seems surprising that a flexible visuomotor mapping takes so long to mature in cataract-treated individuals, since being able to quickly recalibrate the sensorimotor system is essential for everyday adept behavior. Indeed, a flexible sensorimotor system, capable of rapid modification, grants the possibility to rely on fast, feedforward motor control when interacting with the world. Despite the fact that the ability to proficiently recalibrate the visuomotor system keeps developing over the first decade of life or more to reach adult levels (*Bard and Hay, 1983*; *Bard et al., 1990*; *Contreras-Vidal et al., 2005*; *Ferrel et al., 2001*; *Gómez-Moya et al., 2016*; *Hay, 1979*; *Hay et al., 1991*), first signs of this behavior emerge very early after birth in healthy individuals. For example, a few day-old newborns are already able to direct their arm toward a visual target (*von Hofsten, 1982*), and infants can learn new visuomotor transformations to recalibrate the visuomotor system in response to distortions of the visual feedback within the first weeks or months of life (*McDonnell and Abraham, 1979*; *Riddell et al., 1999*). In our study with cataract-treated participants, we did not find equally rapid signs of the emergence of such an ability.

From a comparison between the cataract-treated individuals and the control groups, we can conclude that it is neither age nor the improvement in visual acuity that is the sole determining factor for the development of the recalibration performance, but that such a development requires experience from interaction with the world. First, we can determine that age is not the sole decisive factor by comparing our cataract-treated individual to the group of age-matched sighted controls, which shows that cataract-treated participants are on average much less efficient in reducing the error when exposed to a prismatic shift, and they also present less of an aftereffect following prism removal. In addition, the average rate of recalibration of the group of cataract-treated individuals, who had a mean age of 13 years, is even slower than that of the youngest sighted controls, which was tested at 6 or 7 years of age. Second, we ruled out that the post-surgical visual acuity, which is still lower than that of controls even after cataract removal (*Ganesh et al., 2014*; *Kalia et al., 2014*; *Maurer et al., 2006*), is the only reason for the poorer recalibration behavior by showing that the group of sighted controls with experimentally reduced visual acuity still recalibrates faster than the cataract-treated individuals. This is so even when in general the recalibration rate correlates with the visual acuity after surgery for the cataract-treated individuals. Instead, the performance of the cataract-treated participants improves with time, and thus possibly experience, after surgery, despite the lack of an analogous increase of their visual acuity over time.

Cataract-treated participants differ from sighted controls also in the development of the aftereffect after prism removal. While even the youngest controls reduce the aftereffect trial-by-trial, cataract-treated participants do not present a similar aftereffect decay. Since the pointing task following prism removal is performed in the absence of any visual feedback of the arm, the decline of the aftereffect highlights the contribution of proprioception (*Hamilton and Bossom, 1964*). Indeed, when planning and executing movements toward a visually presented target, healthy individuals rely also on proprioception: they can aim at visually presented targets even if their arm and hand are not in sight, given that there is a stable mapping between the visual and proprioceptive space. The aftereffect decay observed in healthy individuals would indicate that sighted controls tend to

spontaneously return to their normal sensorimotor mapping, based on proprioceptive feedback. The extinction rate of the aftereffect typically increases with children's age, with older children showing a faster decay (*Gómez-Moya et al., 2016*). This is a sign that sighted children learn to rely more and more on proprioception with age (*von Hofsten and Rösblad, 1988*). In contrast, cataract-treated participants do not show the same tendency to quickly reinstate the original mapping. The fact that there is no significant change of the aftereffect with time could either mean that they are simply much slower in reintroducing the original mapping, or that they are unable to integrate proprioceptive information for doing so.

Given the present findings, what could explain the recalibration performance of our cataract-treated participants? To successfully perform the task, they need to reduce the systematic error induced by the prism goggles, while dealing with more variable errors. Such variable error is increased in the cataract-treated participants compared with controls, as can be seen from the noisier performance already at baseline. However, as with time the cataract-treated individuals start to recalibrate faster, while there is no change in the sensorimotor noise, this sensorimotor noise cannot be the only determining factor for the diminished adaptation performance.

Visuomotor recalibration to systematic errors requires participants to establish correspondence between the different sensory systems involved (vision, proprioception) and between the sensory and motor spaces. It could be hypothesized that children who were deprived from pattern vision for the first years of life either lack such sensorimotor mappings, or cannot properly recalibrate them (*Held, 2009*). The present results show that the cataract-treated participants are able to accurately localize and reach for targets quickly after surgery in the absence of any visual distortion, although they do so with greater uncertainty. Some participants are able to point toward targets even before surgery, and therefore participants have either developed a sensorimotor mapping before surgery, or they were able to develop one quickly after surgery. However, despite having developed some form of visuomotor mapping, they were much less able than sighted controls to recalibrate distortions in the mapping once disturbed by prisms. Thus, the main problem seems to be in recalibrating such mappings when perturbed.

To recalibrate such mappings once perturbed, participants need to be able to use an error signal following the introduction of the visual distortion, which is the difference between the location of the visual target and the sensed terminal hand location. The sensed terminal hand location is based on vision as well as on proprioception, and the visual distortion introduces a spatial discrepancy between vision and proprioception as well. Therefore, minimizing the error implies changes in the visuomotor and in the visual-proprioceptive mapping. One of the problems for cataract-treated participants to use such error signals for recalibration might therefore originate from the difficulty to establish correspondence between the visual and motor space, as well as between the visual and proprioceptive space. We recently observed an analogous problem in the development of multisensory integration after cataract removal surgery (*Senna et al., 2021*). Integrating signals from different senses also requires establishing correspondence between different sensory maps and thus may pose similar challenges (*Ernst, 2008*; *Held, 2009*; *Held et al., 2011*). In case of multisensory integration, we have shown that cataract-treated participants–despite other deficits (e.g. *Putzar et al., 2007*; *Guerreiro et al., 2015*)–can learn to optimally integrate multisensory signals within a few years, following a time course analogous to the one observed here.

So far, the possible relationship between integration and recalibration has been the subject of speculation but could never be investigated jointly in the typically developing population. While signs of each of the two abilities emerge early in life, both abilities take years to fully mature (e.g. *Contreras-Vidal et al., 2005*; *Gori et al., 2008*; *Hay, 1990*; *Hay et al., 1991*; *Nardini et al., 2008*), making the study of their developmental path difficult to explore within the same participants over time. Here we had the unique chance to test a subset of the cataract-treated individuals in both studies, the multisensory integration study (*Senna et al., 2021*) and this recalibration study at around the same time after surgery, allowing us to investigate the relationship between these two abilities as they develop after surgery. Strikingly, the performance in the two tasks was significantly correlated: participants who develop better post-surgical multisensory integration abilities in *Senna et al., 2021* also show a better recalibration performance in the present study. Although this is just correlational evidence, it suggests that both tasks are indeed related and that it may be the ability to establish correspondence between the sensory and the sensorimotor maps that undergoes development.

Importantly, with time after surgery cataract-treated individuals are able to learn to recalibrate their visuomotor system, even approaching the performance level of typically sighted participants. This finding indicates that visual experience from pattern vision is necessary for the development of the ability to recalibrate the visuomotor system, which does not mature without pattern vision and thus exposure to sensorimotor distortions. It has been suggested that sensorimotor experience has a pivotal role in the development of the recalibration ability in typically developing children. During development the internal model for such flexible sensorimotor transformations would be learned through experience, via repeated exposure to the sensory consequences of self-generated movements early in life (e.g. *Bauer and Held, 1975*; *Bullock et al., 1993*; *Guigon and Baraduc, 2002*; *Held and Bauer, 1967*). In particular, the recurrent and simultaneous exposure to proprioceptive and visual feedback while executing movements would be used to establish the correspondence between the visual space and the motor space, and between the visual and proprioceptive spaces (*von Hofsten and Rösblad, 1988*).

To summarize, the present study demonstrates that the lack of pattern visual and fine visuomotor experience at an early age affects the ability of cataract-treated individuals to develop flexible sensorimotor mappings. However, the ability to recalibrate the sensorimotor system shows clear improvement over time following cataract removal, even reaching the level of controls tested with visual blur in some cases. The fact that recalibration performance in cataract-treated individuals improves with time after surgery suggests that sensorimotor experience is central in the development of flexible sensorimotor maps. The correlation between the development of multisensory integration and sensorimotor recalibration abilities may hint at the fact that the bottleneck for the development may be in establishing correspondence between the sensory and motor maps.

Being able to use vision to skillfully guide actions requires a well calibrated system and is probably the most important aspect for adept behavior, which we here show is still able to develop with sufficient experience even after many years of visual impairment.

# Materials and methods
## Recalibration in cataract-treated participants and sighted controls
### Participants

Twenty Ethiopian cataract-treated children and adolescents (mean age: 13 years and 2 months, age range: 8–20 years, 19 right-handed, mean time since surgery: 1 year and 8 months, range: 1 day-10 years, mean pre-surgical visual acuity: 1.37 cycles per degree, cpd, range 0.06–3.40 cpd, mean post-surgical visual acuity: 5.04 cpd, range: 1.30–13.45 cpd) took part in the study (see *Figure 1—source data 1* for details). Participants with this condition are extremely rare, therefore the sample size was determined by the availability of individuals suffering from this condition: we tested all the available participants we could find over a period of 3 years (N=20). They presented dense bilateral cataracts, either mature or feremature, or else partially absorbed. Cataracts were classified as congenital, meaning they were either present at birth or developed within the very first few months of life (*Wu et al., 2016*). Such diagnosis was based on the fact that all participants showed optical nystagmus, which is considered a signature of early onset visual deprivation (*Papageorgiou et al., 2014*), and their families reported that children had bright white eyes since birth. Moreover, almost half of the participants had a positive family history of congenital cataract (autosomal dominant), either to one parent or older siblings, suggesting cataracts were hereditary. Furthermore, most participants had misaligned eyes (strabismus) and some other signs suggestive of congenital cataract, such as micro cornea or partially absorbed cataract. They underwent a complete ophthalmological evaluation including B-scan ultra-sound to assure the retina was intact. Inclusion criteria were isolated congenital bilateral cataracts without any other ocular or systemic comorbidity. Participants received ophthalmological evaluation and underwent bilateral cataract surgery and intraocular lens implantation at the Hawassa Referral Hospital, Ethiopia. The target refraction was adjusted for far vision. On average participants were surgically treated 11 years and 5 months after birth (range: 5–19 years). After cataract removal, their vision was still poorer than the normative range (*Kalia et al., 2014*), which is a typical outcome of late surgical treatment (*Carlson and Hyvärinen, 1983*; *Ganesh et al., 2014*; *Hadad et al., 2012*; *Kalia et al., 2014*; *Lewis and Maurer, 2005*; *Maurer et al., 2006*; *Ostrovsky et al., 2006*; *Ostrovsky et al., 2009*, *Figure 1—source data 1*).

We compared the performance of the 20 cataract-treated participants to that of two control groups. The first control group consisted of 20 typically developing sighted German participants (mean age: 13 years and 3 months, age range: 8–19 years and a half, normal or corrected to normal vision, 19 right-handed), individually matched to each cataract-treated participant for age. A second group of 20 sighted German individuals (mean age: 13 years and 4 months, age range: 8–20 years, 19 right-handed) viewed the stimuli through a blurring filter, mimicking the poorer visual acuity exhibited by the cataract-treated participant. Thus, each participant of this second control group was matched to a cataract-treated participant not only for age but also for visual acuity (group mean: 5.05, range: 1.49–14) using the procedure described in *Procedure to blur vision in sighted controls* below. Control participants were randomly assigned to either of the two control groups until the needed number of control participants in the appropriate age-range was met for each group.

Ethiopian participants took part in the experiment at the Hawassa Referral Hospital, at the Shashamane Catholic School for the blind, or at the Sebeta Blind School. German participants were recruited in primary and secondary schools and at Ulm University in Germany. The study was carried out in accordance with the Declaration of Helsinki and approved by the ethics committee of the University of Bielefeld (Bielefeld University, ref nr. EUB 2015–139). Participants, or participants' parents or legal guardians in case of minors, gave their written informed consent to participate in the study and have their anonymized data published in a journal article.

## Visual assessment in cataract-treated participants

Participants visual abilities suffering from congenital bilateral cataracts were evaluated prior to treatment and after cataract removal surgery (see *Figure 1—source data 1* for individual details). Before surgery, all participants had light perception and some of them could see hand motion and additionally count fingers at very close distances. We tested their visual acuity by measuring the contrast sensitivity function (CSF) in all participants after surgery and in a subset of 16 before surgery (*McKyton et al., 2018*; *Senna et al., 2021*). The remaining four were not tested before surgery, either because they had too poor visual acuity to be able to perform the CSF test, or because the procedure was not available at the time they were surgically treated. The subtle variability in pre-surgical visual acuity across participants can be partially explained by the fact that long-standing congenital cataracts can be partially absorbed, thus leaving islands of aphakic clear vision whereas totally white cataract enables light perception only.

The post-surgical CSF was assessed always in the same experimental session as the main experiment. In this test, participants saw a series of Gabor patches (sinusoidal gratings of different spatial frequencies and contrast levels with 19.5 cm Gaussian envelope) presented on a 15.6" gamma-corrected computer display (1920x1,080 pixels resolution). Participants rested their head on a chin-rest at 30 cm distance from the display and had to report whether the grating was oriented horizontally or vertically on each trial. Some participants with extremely poor vision were allowed to perform the test at a shorter distance (15–20 cm) since they would have not been able to perform the task otherwise. In the first block, gratings were all presented at 100% contrast. The test started with a grating at the lowest spatial frequency (0.042 cpd = 1 cycle per 512 pixels at 30 cm viewing distance). As long as the participant's response was correct, a grating with the next higher spatial frequency was presented (up to 10.75 cpd = 1 cycle per 2 pixels). When the participant made the first mistake, a staircase procedure was introduced: 3 correct responses in a row led to the next higher frequency, while 1 mistake led to the next lower frequency (i.e. 3 up-1 down staircase). We used a total of 9 spatial frequencies evenly spaced on a logarithmic scale (i.e. 0.042, 0.084, 0.168, 0.336, 0.672, 1.344, 2.688, 5.375, 10.75 cpd). The procedure stopped after 6 reversals. In a second block, each of the spatial frequencies was kept constant while the contrast was varied. The frequencies were tested separately one after the other, from the lowest to the highest, starting one frequency step higher than the spatial threshold frequency assessed in the first block. For each spatial frequency, the first grating was presented at 100% contrast. As long as the participant's responses were corrected, the contrast was gradually reduced (to a minimum of 0.78%). Upon the first error, a 3 up-1 down staircase procedure similar to the one in the first block was used, to measure the participant's contrast threshold at each frequency, calculated as the average contrast of the last six reversals. A total of eight contrast levels (equally spaced logarithmically) were used. For each frequency, we took the logarithm of the sensitivity (1/contrast threshold) and plotted it as a function of spatial frequency (also log-transformed), yielding the

participant's CSF. The CSF was fitted with an inverse parabola (*McKyton et al., 2018*; *Senna et al., 2021*; *Watson and Ahumada, 2005*) to get the CSF cutoff frequency, namely the highest spatial frequency that the participant could still see at the maximal contrast.

## Procedure to blur vision in sighted controls

To investigate whether any possible difference in the performance in the pointing task between cataract-treated and sighted individuals might simply result from the lower visual acuity exhibited by the former, we blurred vision in a group of 20 sighted individuals, to mimic the poor visual acuity that the cataract-treated participants still experienced after surgery. To this end, we placed a transparent Plexiglas panel covered by a blurring transparent plastic foil on top of the setup used during the task (see *Figure 1A*). Changing the distance between the blurring screen and the visual targets varied the amount of blur applied to the visual target, with a greater blurring factor for greater distance. We ran a pilot study to select the range of distances between the screen and the visual target that would be needed in order to reproduce the visual acuity of the cataract-treated participants in terms of experienced blur levels and contrast reduction. Nevertheless, to ensure that this procedure would lead to the desired visual acuity in the control participants, we tested their visual acuity with the same orientation discrimination task used for the cataract-treated participants. The CSF of the control participants was measured by placing the blurring panel at the desired distance from the computer's monitor. We visually inspected the contrast sensitivity function of each control participant and we included only the controls in the study who presented contrast sensitivity functions matching those of the cataract-treated participants for both CSF cutoff frequency and shape (see *Figure 1—figure supplement 1*). This led to the exclusion of 8 control participants, in which we failed to obtain the desired CSF shape and cutoff frequency. Each sighted control of the final sample of 20 participants was individually matched to one cataract-treated participant for both visual acuity and age.

## Experimental procedure

Participants sat at a table, in front of the box-like setup (27 cm high, 76 cm wide, 37 cm deep), placed on the edge of the table. The side of the box proximal to the participants was open, so that participants could place their arms inside. Thus, the upper side of the box hid the participant's arm from sight (see *Figure 1A*). Depending on the length of the participant's arm, the depth of the box could be adjusted, from 37 to 30 cm. Subjects were instructed to repeatedly point toward a visual target (the red cap of a marker, 3.5 cm high, 1.6 cm wide) placed at the distal side of the box. They were asked to point with their dominant hand fast but at a comfortable speed. Participants performed the movements inside the box, and after each pointing they returned their hand to a starting position aligned to their mid-sagittal axis. The target was presented manually by the experimenter at the distal side of the box, right above its edge (around 40–50 cm from the participant's eye). In each trial, the target could be shown at one of three possible locations: straight ahead in front of the subject (0°), 25° to the left or to the right of the participant's body midline. At the back of the box a ruler was attached such that the experimenter could determine and record the participant's pointing location (*Fortis et al., 2010*; *Frassinetti et al., 2002* for a similar procedure). Participants' head was kept aligned with their body's sagittal axis by a chin-rest, and the experimenter made sure that the participant would not move the head during the experiment.

The experiment consisted of three phases. During the *pre-prism* phase, participants performed 12 pointing movements (four for each target location). The far side of the box was closed by means of a removable semi-transparent Plexiglas panel in order to prevent participants from seeing their finger reaching out of the box at the far side. Therefore, the task was performed in the absence of any visual feedback of the hand and finger movement. In this *pre-prism* phase, participants wore plastic goggles without any distorting lens and thus they had a natural view on the scene. In the next phase (*prism*), a prismatic lens was introduced into the goggles, shifting the visual field by 20 prism diopters (i.e. 11.31°) toward the right. In this phase, the Plexiglas panel was removed and subjects performed 48 pointing trials (16 for each target location) with terminal visual feedback of their fingers position. That is, since the movement was executed below the top of the box, participants could only see the tip of their finger emerging from the distal edge of the box (terminal feedback). In this Way, participants could not correct the movement along the way, but only in the next trial based on the terminal pointing error. In the last phase (*post-prism*), the prism lens was removed and the Plexiglas

panel was reintroduced, as in the *pre-prism* phase, and participants performed 48 trials (16 for each target location) again without terminal visual feedback. In each phase, targets were presented in a pseudorandom order, with the same number of trials for each of the three target positions. Overall, the experiment consisted of 108 trials and lasted about 20 min.

As the prolonged absence of pattern vision typically results in amblyopia and in stereoacuity deficits, all participants were tested monocularly. Cataract-treated participants were tested with their better eye based on medical examination and participants' self-report. Sighted subjects were tested with their dominant eye as determined by the hole-in-the-card test (*Durand and Gould, 1910*). Out of all participants, 13 cataract-treated participants, 11 sighted controls tested with normal vision condition, and 10 controls tested with blurred vision were left eye dominant.

## Statistical analyses

Performance in the pointing task was assessed by examining pointing errors, as the difference–in degrees of visual angle–between the recorded pointing position and the location of the target. A negative score indicated a leftward error with respect to the target, while a positive score indicated a rightward error. To quantify error reduction in the *prism* phase, in each group we fitted a power function on the mean pointing errors across the participants of each group across all trials of the *prism* phase ($Error = a * x^b$, see main text). Note that the individual profiles in sighted participants were best described by exponentials. However, due to noise in the individual profiles of the cataract-treated participants, the curves were fitted on the group mean, and the mean of multiple exponentials with different rate parameters typically approximates a power function.

To compare the error made in the *pre-prism* baseline and in the *post-prism* phase across the three groups, we calculated the mean pointing error across all trials of each of the two phases in each participant. For each phase, we compared the mean pointing error across the three groups using a Kruskal–Wallis test, followed by pairwise comparisons carried out with Bonferroni-corrected Wilcoxon rank-sum tests (see main text). Furthermore, we compared pointing precision across the three groups in the *pre-prism* phase by means of Bonferroni-corrected pairwise two-sample F-tests on the variance of the pointing errors. For all analyses, we set the significance level alpha to 0.05. Note that non-parametric tests were used here and elsewhere in the text whenever the normality assumption was violated.

To analyze individual recalibration performance, we calculated a recalibration index ($i_{recal}$) within each participant. This index combined the amount of recalibration in the *prism* and at the beginning of the *post-prism* phases (*Prism Adaptation* and Initial *Aftereffect*, respectively). *Adaptation* was calculated as the error reduction in the *prism* phase (induced *Prism Distortion*, 11.31°, minus *End Prism*, that is average of the last three pointing errors of the *prism* phase, *Fortis et al., 2010*). *Initial Aftereffect* was calculated as the magnitude of the aftereffect exhibited right after prism removal (i.e., average of the first three pointing errors of the *post-prism* phase). $i_{recal}$ was calculated as the average between *Adaptation* and the (negative) *Initial Aftereffect*. The result was normalized to the prism distortion (i.e., 11.31°), leading to an index ranging between 0 and 1, with higher $i_{recal}$ indicating stronger recalibration:

$$i_{recal} = \frac{1}{Prism\ Distortion} \frac{Adaptation - Initial\ Aftereffect}{2}, \tag{3}$$

In detail:

$$i_{recal} = \frac{1}{Prism\ Distortion} \frac{(Prism\ Distortion - End\ Prism) - Initial\ Aftereffect}{2}. \tag{4}$$

Note that only the initial phase of the aftereffect (first three trials) contributes to $i_{recal}$. The *Initial Aftereffect* allows us to appreciate the magnitude of the error right after the prism goggles are removed (i.e. the initial systematic error in the *post-prism* phase). However, it does not allow us to appreciate the development of the aftereffect during the whole *post-prism* phase. The *post-prism* phase differs substantially from the *prism* phase, not only because the prisms are removed, but also because pointing is performed in the absence of the terminal visual feedback (and therefore, in the *post-prism* phase it is possible to mainly appreciate the contribution of proprioception to the development of the aftereffect). For this reason, the recalibration index $i_{recal}$ does not include also the development of the aftereffect in the whole *post-prism* phase, but instead includes only the initial three trials.

Detailed analyses on the aftereffect during the entire *post-prism* phase are reported in *Figure 1—figure supplement 5*, where we calculated the aftereffect as the mean pointing error across all trials of the *post-prism* phase. Furthermore, as previous evidence has shown that the aftereffect tends to gradually decay (*Hamilton and Bossom, 1964*), we investigated its trial-by-trial temporal unfolding in each group (for more details, see *Figure 1—figure supplement 4*).

In cataract-treated participants, we correlated $i_{recal}$ with the variance of the pointing errors in the *pre-prism* phase and with visual acuity (i.e. the log-transformed CSF cutoff frequency) via Pearson's correlation coefficient. Outliers, defined as values three standard deviations from the mean, were excluded from the analysis. This led to excluding one participant from the analyses on the variance of the error in the *pre-prism* phase.

In *Figure 1—figure supplement 5*, we provided the same correlational analyses, but conducted separately for *Adaptation* and *Aftereffect* (calculated as mean pointing error across all trials of the whole *post-prism* phase). *Adaptation* and *Aftereffect* were significantly correlated.

Analyses were performed with MATLAB.

## The contribution of time after surgery to recalibration

### Participants

We had the chance to test a subtest of 13 participants over time. They were all tested twice after surgery: the first post-surgical test took place between 1 day and 2 years and 4 months after surgery (mean time since surgery: 6 months, mean visual acuity: 4.97 cpd, range: 1.30–13.04 cpd). The follow up test took place between 4 months and 1 year and 4 months after the first test (mean: 11 months and a half; visual acuity: 6.19 cpd, range: 1.35–13.72 cpd, *Figure 1—source data 1* for further details). Note that 4 participants had enough residual vision to be able to see the target and perform the task even before surgery, and were therefore tested also prior to the operation (see *Figure 1—source data 1* for details).

### Procedure and statistical analyses

Participants took part in the same recalibration task twice. We also tested participants' visual acuity in each experimental session. We compared participants' performance in the two sessions, by calculating their $i_{recal}$ in each session and comparing the two via a Wilcoxon signed-rank test.

## Relations between recalibration and multisensory integration

### Participants

A subset of 14 out of the 20 cataract-treated participants were also tested previously in a study on multisensory perception which assessed the ability to integrate vision with touch after cataract removal (*Senna et al., 2021*). Participants took part in the present and in the previous study at around the same time after surgery (mean age: 13 years and 8 months, range: 8–19 years, 13 right-handed; recalibration task, mean time since surgery: 1 years and 11 months, range: 1 day-10 years and 5 months, mean visual acuity: 4.34 cpd, range: 1.30–12.14 cpd; multisensory task, mean time since surgery: 1 year and 10months, range: 2 days-10 years and 5 months, mean visual acuity: 4.38 cpd, range: 1.30–13.87 cpd).

### Procedure of the multisensory integration study

In the study by *Senna et al., 2021*, we investigated whether cataract-treated participants can combine information from vision and touch. We asked them to match the size of an object (standard) explored visually (V), haptically (H) or visual-haptically (VH) to one of a series of ten objects differing in size (comparisons), presented either visually or haptically. Thus, there were two unisensory conditions (i.e. V-V: standard and comparison both presented visually, and H-H: both presented haptically) and two multisensory conditions (i.e. VH-V: standard presented visual-haptically and comparison presented visually, and VH-H: standard presented visual-haptically and comparison haptically). To assess the influence of an estimate in one modality onto another, we introduced a discrepancy between the senses: the object was observed through a magnifying lens, so that it looked bigger than it was perceived with touch. By comparing the two multisensory size estimates (VH-V and VH-H), it is possible to estimate the mutual influence (*Multisensory Influence, MI*) of vision on touch and vice versa. If participants do not integrate the multisensory signals the Multisensory Influence is MI = 0. In case the signals

get fused into a singular estimate, the Multisensory Influence is *MI = 1*. Thus, *MI* is a measure of the strength of the multisensory influence and hence can be used as a signal for the development of multisensory integration.

## Statistical analyses

We assessed whether the performance in the recalibration task (present study) and the multisensory integration task (*Senna et al., 2021*) were related by correlating $i_{recal}$, as a measure of recalibration performance and the Multisensory Influence (*MI*) as a measure of integration performance via Pearson's correlation coefficient.

# Acknowledgements

We were grateful to Ehud Zohary and Ayelet McKyton for organizing surgeries and stays in Ethiopia. We were grateful to Zemene Zeleke for coordinating and organizing our testing in Ethiopia and to Julia Mies and Theresa Lask for helping with data collection in Germany. This study was supported by the Deutsche Forschungsgemeinschaft (DFG) German-Israel cooperation DIP-Grant awarded to MOE (ER 542/3–1).

# Additional information

## Funding

| Funder | Grant reference number | Author |
| --- | --- | --- |
| Deutsche Forschungsgemeinschaft | ER 542/3-1 | Marc O Ernst |

The funders had no role in study design, data collection and interpretation, or the decision to submit the work for publication.

## Author contributions

Irene Senna, Conceptualization, Data curation, Formal analysis, Investigation, Visualization, Methodology, Writing – original draft; Sophia Piller, Investigation, Writing - review and editing; Itay Ben-Zion, Investigation, Writing - review and editing, Responsible for medical and optometric examinations, surgeries, and supervision of the local Ethiopian medical staff; Marc O Ernst, Conceptualization, Supervision, Funding acquisition, Methodology, Writing - review and editing

## Author ORCIDs

Irene Senna (iD) http://orcid.org/0000-0002-3237-8193

## Ethics

The study was carried out in accordance with the Declaration of Helsinki and approved by the ethics committee of the University of Bielefeld (Bielefeld University, ref nr. EUB 2015-139). Participants, or participants' parents or legal guardians in case of minors, gave their written informed consent to participate in the study and have their anonymized data published in a journal article in an anonymous form.

## Decision letter and Author response

Decision letter https://doi.org/10.7554/eLife.78734.sa1
Author response https://doi.org/10.7554/eLife.78734.sa2

# Additional files

## Supplementary files

• MDAR checklist

## Data availability

The full dataset including all the experimental results and the participants' demographic information has been deposited on Mendeley: https://doi.org/10.17632/ksdwxdwtxg.2.

The following dataset was generated:

| Author(s) | Year | Dataset title | Dataset URL | Database and Identifier |
|---|---|---|---|---|
| Senna I | 2022 | Recalibrating vision-for-action requires years after sight restoration from congenital blindness | https://doi.org/10.17632/ksdwxdwtxg.2 | Mendeley, 10.17632/ksdwxdwtxg.2 |

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
