## [Editor Report]

This valuable paper will be of interest to researchers in the fields of motor control, visual perception, learning and brain plasticity, sight loss, and rehabilitation. The paper shows the contributions of sensory-motor experience to the development of visuo-motor recalibration abilities using convincing and careful experimental methods and analyses, comparing a rare population of late-operated cataract patients with normal-sighted control groups.

---

## [Decision Letter]

**Decision letter after peer review:**

Thank you for submitting your article "Recalibrating vision-for-action requires years after sight restoration from congenital blindness" for consideration by *eLife*. Your article has been reviewed by 2 peer reviewers, and the evaluation has been overseen by Miriam Spering as the Reviewing Editor and Chris Baker as the Senior Editor. The following individual involved in review of your submission has agreed to reveal their identity: Marko Nardini (Reviewer #1).

Essential revisions:

1) Interpretation of baseline precision pointing results. Reviewer 1 suggests alternative explanations that could be ruled out with additional analyses (e.g., if initial pointing variability predicts recalibration) or collection of additional perturbation data in the control group only such that the baseline data are better matched to the patient group.

2) CSF matching. It is unclear what exactly was achieved with the blurring manipulation. The authors need to provide additional detail here and address this point raised by Reviewer 1 carefully.

3) Data analysis. Reviewer 2 suggests analyzing adaptation and aftereffects separately throughout.

4) Please pay attention to how negative findings are described in this manuscript. The absence of a significant statistical effect does not provide evidence for the absence of the effect. This problem arises at several instances throughout the paper. Please outline in detail how this issue was addressed.

5) Writing and logical structure. Please be sure to report Results in this dedicated section, not in the Discussion, and explain Methods clearly and succinctly in methods, not in the Results (see Reviewer 2 comment on structure).

Please see below for additional comments, and please submit a point-by-point reply to all of them.

*Reviewer #1 (Recommendations for the authors):*

(1) Consideration of the role of greater baseline pointing variability in patients.

(2) More detail on the CSF.

*Reviewer #2 (Recommendations for the authors):*

In instances, the results should be described more neutrally. For example, "higher visual acuity and thus a better visual input leading to better quality experience led to better recalibration performance" (l. 192). That the association between higher visual acuity and faster visuo-motor recalibration indeed was mediated by 'better quality experience' is by no means warranted, for example, higher visual acuity also implies more precise visual feedback during the adaptation phase. In a similar vein, it might be more realistic to speak of observations or findings rather than facts, unless an effect has been replicated in a number of different studies.

The manuscript would profit from more structure. For example, it is confusing that the existence of repeated measurements from multiple participants is first mentioned within a paragraph discussing the relationship between the time since surgery and the amount of visuo-motor recalibration. Similarly, the results of an analysis comparing the recalibration and multisensory integration in the same sample are reported in the discussion rather than in the Results section. As another example, it is confusing to learn about several of the control experiments in the participants section of the methods.

[Editors’ note: further revisions were suggested prior to acceptance, as described below.]

Thank you for resubmitting your work entitled "Recalibrating vision-for-action requires years after sight restoration from congenital blindness" for further consideration by *eLife*. Your revised article has been evaluated by Chris Baker (Senior Editor) and Miriam Spering (Reviewing Editor).

The manuscript has been improved but there are still some substantive issues that need to be addressed, as outlined below:

The impression here is that you bury an interesting result in a manuscript that is overall not yet well-structured and overly wordy, contains too much detail that could easily be redacted or summarized in table form, and even some results that could be entirely omitted. Please see reviewer 2's comments for details. I would also like to emphasize that it is critically important that all statistical errors and omissions be fixed at this point, in particular with reporting t-tests and correctly applying linear fits and F-tests. Please include a point-by-point reply to reviewer 2's remaining issues with your resubmission.

*Reviewer #1 (Recommendations for the authors):*

I appreciate the authors' detailed responses and additions to the MS (consideration of different pointing variability baselines, full CSFs, details of index calculation, more minor points), which deal satisfactorily with these issues. The MS as it now stands reads to me as a clear and significant contribution to the literature.

*Reviewer #2 (Recommendations for the authors):*

The authors did a great job dissociating adaptation effects from aftereffects and from recalibration based on proprioception alone in the revised version of the manuscript. The newly introduced information about pointing variability is very interesting.

There remains some confusion about the control participants. First, there is one German group and then another one, each one matched to the cataract patients. In the next paragraph then there are Ethiopian groups matched to the German groups rather than to the cataract patients. The latter groups do not differ in their pointing profiles, but it is not clear what is meant by pointing profiles and how these were compared. Most importantly, it remains unclear whether German and Ethiopian controls were combined for the analyses and figures shown in the main paper. If so and if I understand correctly, group sizes would differ which makes SEM error bars hard to interpret. In turn, not showing data of all tested groups seems highly problematic.

It makes no sense to fit a linear regression line to a non-linear process. In linear regression, the influence of data points on the estimated coefficients increases with their distance from the mean value of the predictor. Adding a confidence band to the predicted growth curve and discussing the confidence interval of the estimate for reaching asymptotic values as it is already done in most parts of the revised manuscript are the best ways to transport the message that it takes between months and years to recover the ability for visuomotor recalibration.

The title still contains the word blindness, this will mislead many readers given that the participants had residual light perception.

Every report of a statistical test needs to include the value of the test statistic, p-values alone are not sufficient and render it impossible to conduct meta-analyses.

Comparative claims, e.g., 'much faster' (174) should be supported by a statistical test.

Variances are typically compared with an F-test.

The manuscript is still a cumbersome read, I needed several runs to not be lost. Several easy changes could greatly improve the readability of the manuscript.

– Headers (if necessary, in the paragraph style employed in parts of the Methods section) could guide the reader and make it much easier to sort the results mentally.

– Order based on importance, content, and trustworthiness of the results.

– The repeated testing of cataract individuals is one of the core elements of the paper and is extremely interesting. Yet, the results are tugged away after a huge set of correlations, which are far less interesting as the power far too low for any reliable correlational analysis.

– An experiment testing the effects of age with 12 participants raises doubts.

– Experimental details, such as the procedure to match the visual blur, monocular testing, or the power function equation, might be better suited for the methods section.

– All results should be reported in one section of the paper and for every included result the experiment and its purpose should be described properly.

– It seems as if some results (e.g., those of the dissipation of the aftereffect, i.e., recalibration based on proprioception alone) are only described in the figure caption, which will fill several pages.

– Other results, e.g., the longitudinal study in sighted individuals, are mentioned as side notes but neither methods nor results are described in detail.

– Some results are still not described in a fashion that allows the reader to form a judgment. For example, 'multisensory influence' might mean very different things depending on the paradigm and the parameter that has been calculated. readers should not be asked to go to another paper to find out.

– There are still results that only occur in the Methods section, which should not contain results.

– A lot of text could be removed from the Methods section by introducing a table with each participant's data and checkmarks indicating which tests they took part in. This would also provide an overview over the different tests.

– Exclusions should be described in the participants section or if a participant's data were only excluded from a single analysis, exclusions should be described at the beginning of that section.

– It might make more sense to structure the Methods section by experiment than describing all experiments at once even once the results are reported more orderly.

– If *eLife* allows doing so, it could be very useful to move several analyses (incl. the respective methods and results) and even some additional experiments into the supplement. A reader has to work really hard to extract the main result (reduced recalibration that recovers over time) from the flurry of information, which is a shame because the result is so interesting.

– The discussion is very long and speculative. Less might well be more.

Figures

For each plot showing group averages, the number of participants per group should be indicated in the figure.

Figures 1C, D, F, G, S1, S3, S4 confidence for the estimated lines/curves would be very helpful to judge the uncertainty associated with the provided estimates.

1A Was the blurring shield shown in the figure used to prevent vision of the endpoint at the end of the movement or is this the shield used to degrade sighted participants' vision.

Please clarify in the caption or the label. Labeling the pointing target as such would be helpful if a reader looks at the figure before reading the text.

1B Does the figure show mean pointing errors of individual participants as indicated in the caption or does it show group averages of these participant-level averages?

1C Pre Phase – Pre-prism phase and the word should be moved to the x-axis because only the variance refers to the pre-prism phase only.

1F a legend (red: first test, brown: second test) would be helpful.

S4 it would be helpful to indicate the correlation coefficient and its p-value.

---

## [Author Response]

Reviewer #1 (Recommendations for the authors):(1) Consideration of the role of greater baseline pointing variability in patients.

Thank you, we have now added this important analysis.

(2) More detail on the CSF.

We have now provided more information and a figure on the CSF in Material and Methods and in a supplemental figure.

Reviewer #2 (Recommendations for the authors):In instances, the results should be described more neutrally. For example, "higher visual acuity and thus a better visual input leading to better quality experience led to better recalibration performance" (l. 192). That the association between higher visual acuity and faster visuo-motor recalibration indeed was mediated by 'better quality experience' is by no means warranted, for example, higher visual acuity also implies more precise visual feedback during the adaptation phase. In a similar vein, it might be more realistic to speak of observations or findings rather than facts, unless an effect has been replicated in a number of different studies.

We have removed these parts and rephrased the text more carefully.

The manuscript would profit from more structure. For example, it is confusing that the existence of repeated measurements from multiple participants is first mentioned within a paragraph discussing the relationship between the time since surgery and the amount of visuo-motor recalibration. Similarly, the results of an analysis comparing the recalibration and multisensory integration in the same sample are reported in the discussion rather than in the Results section. As another example, it is confusing to learn about several of the control experiments in the participants section of the methods.

We have modified the structure and rephrased these parts, as suggested by Reviewer 2.

[Editors’ note: what follows is the authors’ response to the second round of review.]

The impression here is that you bury an interesting result in a manuscript that is overall not yet well-structured and overly wordy, contains too much detail that could easily be redacted or summarized in table form, and even some results that could be entirely omitted. Please see reviewer 2's comments for details. I would also like to emphasize that it is critically important that all statistical errors and omissions be fixed at this point, in particular with reporting t-tests and correctly applying linear fits and F-tests. Please include a point-by-point reply to reviewer 2's remaining issues with your resubmission.

Following the suggestions of Reviewer 2 and the recommendation of the editor, we have shortened the manuscript, omitted results of secondary importance, and better structured the manuscript with the help of headers. Furthermore, we have fixed the statistical omissions.

Reviewer #2 (Recommendations for the authors):The authors did a great job dissociating adaptation effects from aftereffects and from recalibration based on proprioception alone in the revised version of the manuscript. The newly introduced information about pointing variability is very interesting.

We thank Reviewer 2 for appreciating our work and for the suggestions that helped to further improve our manuscript.

There remains some confusion about the control participants. First, there is one German group and then another one, each one matched to the cataract patients. In the next paragraph then there are Ethiopian groups matched to the German groups rather than to the cataract patients. The latter groups do not differ in their pointing profiles, but it is not clear what is meant by pointing profiles and how these were compared. Most importantly, it remains unclear whether German and Ethiopian controls were combined for the analyses and figures shown in the main paper. If so and if I understand correctly, group sizes would differ which makes SEM error bars hard to interpret. In turn, not showing data of all tested groups seems highly problematic.

We agree with Reviewer 2 that the description of the participants’ groups in the different experiments may have been confusing. In the previous version of the manuscript, we compared the Ethiopian and German sighted controls via the same analyses that were used in the main text to compare cataract-treated participants to the sighted controls. In other words, we made sure that the Ethiopian controls did not differ from the German control groups in accuracy and precision in the *pre-prism* phase, in the error reduction in the *prism* phase, in the recalibration index *i_recal_*, et cetera. We did not combine the Ethiopian and German groups for the analyses. Testing also a few Ethiopian controls was done only as a sanity check (to exclude that any cultural effects of some sort could affect performance). However, given that including these participants might be a source of confusion (as it may add complexity, without providing essential information) we have now removed that section. Indeed, we have now removed some analyses of secondary importance and omitted some experiments from this version of the manuscript to improve clarity (see further points below). However, in case the Reviewer thinks otherwise, we will be happy to reconsider our choice and provide an additional supplemental figure with the comparisons across German and Ethiopian controls.

It makes no sense to fit a linear regression line to a non-linear process. In linear regression, the influence of data points on the estimated coefficients increases with their distance from the mean value of the predictor. Adding a confidence band to the predicted growth curve and discussing the confidence interval of the estimate for reaching asymptotic values as it is already done in most parts of the revised manuscript are the best ways to transport the message that it takes between months and years to recover the ability for visuomotor recalibration.

We have followed the suggestions of Reviewer 2 and removed the linear fit on the time since surgery, leaving only the exponential fit in the text (Results section, lines 325-369 and Figure 1).

The title still contains the word blindness, this will mislead many readers given that the participants had residual light perception.

We thank Reviewer 2 for drawing our attention to this point. We have now removed the term ‘blindness’ and introduced ‘cataracts’ instead.

Every report of a statistical test needs to include the value of the test statistic, p-values alone are not sufficient and render it impossible to conduct meta-analyses.

We have now fixed these omissions.

Comparative claims, e.g., 'much faster' (174) should be supported by a statistical test.

We have rephrased that sentence and we now talk about differences in average and non-overlapping confidence intervals (lines 164-165).

Variances are typically compared with an F-test.

We have now compared variances with F-tests (main text, lines 135-137, methods 908-909 and legend of Figure supplement 2).

The manuscript is still a cumbersome read, I needed several runs to not be lost. Several easy changes could greatly improve the readability of the manuscript.– Headers (if necessary, in the paragraph style employed in parts of the Methods section) could guide the reader and make it much easier to sort the results mentally.– Order based on importance, content, and trustworthiness of the results.

We thank the Reviewers for these suggestions that have helped improve the readability of the manuscript. We used headers to structure the Results and Materials and methods sections, we modified the order in which we present some results, omitted some results, or moved them to the Supplementary material.

– The repeated testing of cataract individuals is one of the core elements of the paper and is extremely interesting. Yet, the results are tugged away after a huge set of correlations, which are far less interesting as the power far too low for any reliable correlational analysis.

We have now removed most of the correlational analyses and moved their description into the Material and Methods section and Supplementary Materials. The repeated testing is now highlighted in the paragraph ‘The contribution of time after surgery to recalibration’, in the Results section (from line 215).

– An experiment testing the effects of age with 12 participants raises doubts.

We have removed this analysis.

– Experimental details, such as the procedure to match the visual blur, monocular testing, or the power function equation, might be better suited for the methods section.

We have moved some procedural details into the methods section, but left details that will help the reader understand procedures and analyses. For instance, we decided to leave the power function in the text, as such explanation is important for understanding the results on the temporal parameter *b*, which otherwise may remain obscure. Similarly, we kept a short description of the procedure used to match the CSF.

– All results should be reported in one section of the paper and for every included result the experiment and its purpose should be described properly.– It seems as if some results (e.g., those of the dissipation of the aftereffect, i.e., recalibration based on proprioception alone) are only described in the figure caption, which will fill several pages.

It is our understanding from the guidelines of the Journal, that figure supplements can be used to provide additional analyses or data linked to primary figures, with information about data processing and analyses provided in the figure legends. Following these guidelines, we have provided additional details on the analyses in the supplementary figures and relative legends that are not reported in the Results section of the main text. However, we agree with the Reviewer that some figure captions were very long. For this reason, we have now reduced the complexity of the supplementary figures: we have split Figure Supplement 2 (previously consisting of two panels) into 2 different figures and we have removed one of the two panels of Figure supplement 6 (i.e., entirely omitting the additional experiment on the effect of visual blur in adults).

– Other results, e.g., the longitudinal study in sighted individuals, are mentioned as side notes but neither methods nor results are described in detail.

The longitudinal study is mentioned in the main text (lines 119-120 and 217), but presented in detail only in the Supplementary material, see Figure supplement 6 (in line with one suggestion of the Reviewer in a point below).

We have now entirely removed the study on the effect of blurring vision in controls: Again, this would add complexity, without adding essential information to the story.

– Some results are still not described in a fashion that allows the reader to form a judgment. For example, 'multisensory influence' might mean very different things depending on the paradigm and the parameter that has been calculated. readers should not be asked to go to another paper to find out.

We have now added a paragraph in the methods section (entitled ‘Procedure of the multisensory integration study’, from line 1001) describing the multisensory task and the ‘multisensory influence’ measure.

– There are still results that only occur in the Methods section, which should not contain results.

We have now moved these results into the Results section and/or in the figure legends.

– A lot of text could be removed from the Methods section by introducing a table with each participant's data and checkmarks indicating which tests they took part in. This would also provide an overview over the different tests.

Individual data are reported for each participant in a table in the Supplementary material (Figure 1_Source data 1). However, we chose to report their average demographics also in the methods section for clarity and completeness. However, we have shortened and re-arranged the methods section to improve clarity.

– Exclusions should be described in the participants section or if a participant's data were only excluded from a single analysis, exclusions should be described at the beginning of that section.

We have now fixed this point.

– It might make more sense to structure the Methods section by experiment than describing all experiments at once even once the results are reported more orderly.

We have followed the Reviewer’s suggestion, which we believed helped clarity.

– If eLife allows doing so, it could be very useful to move several analyses (incl. the respective methods and results) and even some additional experiments into the supplement. A reader has to work really hard to extract the main result (reduced recalibration that recovers over time) from the flurry of information, which is a shame because the result is so interesting.

It is our understanding from the guidelines of the Journal, that figure supplements can serve this purpose. We have removed some non-essential detail or analysis, however, we have kept additional analyses in the supplementary figures and respective legends (e.g., the longitudinal data are mentioned in the main text, but shown in detail only in the supplementary material, i.e., as a suppl. figure). However, if the Editor suggests otherwise, we will be happy to provide some supplemental material in another form.

– The discussion is very long and speculative. Less might well be more.

We have now shortened the discussion.

FiguresFor each plot showing group averages, the number of participants per group should be indicated in the figure.

We now reported the number of participants for each plot in the figure legend.

Figures 1C, D, F, G, S1, S3, S4 confidence for the estimated lines/curves would be very helpful to judge the uncertainty associated with the provided estimates.

We have now added the 95% confidence intervals and visualized, where helpful, the uncertainty with the estimates in the figures.

1A Was the blurring shield shown in the figure used to prevent vision of the endpoint at the end of the movement or is this the shield used to degrade sighted participants' vision.Please clarify in the caption or the label.

The shield was used to blur controls’ vision, while the cover was used to prevent vision of the terminal feedback. We have better specified it in the figure legend of Figure 1.

Labeling the pointing target as such would be helpful if a reader looks at the figure before reading the text.

We have now inserted a label for the target in the figure.

1B Does the figure show mean pointing errors of individual participants as indicated in the caption or does it show group averages of these participant-level averages?

We have clarified this point in the legend of Figure 1: we calculated mean pointing errors across bins of three trials for each participant and we then show group averages for the three groups.

1C Pre Phase – Pre-prism phase and the word should be moved to the x-axis because only the variance refers to the pre-prism phase only.

We have modified the figure.

1F a legend (red: first test, brown: second test) would be helpful.

We have added this legend to the figure.

S4 it would be helpful to indicate the correlation coefficient and its p-value.

We have reported correlation coefficients and p-values in the figure legend.